# Clinical Utility of the EpiSwitch CiRT Test to Guide Immunotherapy Across Solid Tumors: Interim Results from the PROWES Study

**DOI:** 10.3390/cancers17172900

**Published:** 2025-09-04

**Authors:** Joe Abdo, Joos Berghausen, Ryan Mathis, Thomas Guiel, Ewan Hunter, Robert Heaton, Alexandre Akoulitchev, Sashi Naidu, Kashyap Patel

**Affiliations:** 1Oxford BioDynamics Inc., 7495 New Horizon Way, STE 110, Frederick, MD 21703, USA; joe.abdo@oxfordbiodynamics.com (J.A.); rmathis04@gmail.com (R.M.); 2Department of Pharmacology and Physiology, Georgetown University Medical Center, 3900 Reservoir Rd NW, Washington, DC 20057, USA; jhb109@georgetown.edu; 3Oxford BioDynamics CLIA Laboratory, 7495 New Horizon Way, STE 110, Frederick, MD 21703, USA; thomas.guiel@oxfordbiodynamics.com (T.G.); robert.heaton@oxfordbiodynamics.com (R.H.); 4Oxford BioDynamics Plc., 3140 Rowan Palace, John Smith Drive, Oxford OX4 2WB, UK; ewan.hunter@oxfordbiodynamics.com (E.H.); alexandre.akoulitchev@oxfordbiodynamics.com (A.A.); 5Carolina Blood & Cancer Care Associates, 1583 Healthcare Drive Rock Hill, Rock Hill, SC 29732, USA; snaidu@cbcca.net

**Keywords:** checkpoint inhibitors, EpiSwitch CiRT assay, immunotherapy response, 3D genome conformation, blood-based cancer diagnostics, predicting checkpoint inhibitor response, real-world evidence, precision oncology

## Abstract

**Simple Summary:**

The EpiSwitch^®^ CiRT blood test helps oncologists determine which patients are most likely to benefit from immune checkpoint inhibitor (ICI) therapy. By guiding treatment decisions, it reduces unnecessary exposure to ineffective or costly treatments, minimizes patient harm, and supports personalized cancer care. Real-world data support its value as a non-invasive, cost-efficient tool for optimizing immunotherapy selection across cancer types.

**Abstract:**

**Background:** Immunotherapy has revolutionized oncology care, but clinical response to immune checkpoint inhibitors (ICIs) remains unpredictable, and treatment carries substantial risks and costs. The EpiSwitch^®^ CiRT blood test is a novel 3D genomic assay that stratifies patients by probability of ICI benefit using a binary, blood-based classification: high (HPRR) or low (LPRR) probability of response. **Methods:** This interim analysis of the ongoing PROWES prospective real-world evidence study evaluates the clinical utility of CiRT in 205 patients with advanced solid tumors. The primary endpoint was treatment decision impact, assessed by pre-/post-test physician surveys. Secondary endpoints included treatment avoidance, time to ICI initiation, concordance with clinical response, early discontinuation rates, and exploratory health economic modeling. Longitudinal use, resistance monitoring, and equity analysis by social determinants of health (SDoH) were also explored. **Results:** CiRT results influenced clinical decision-making in a majority of cases. LPRR status was associated with higher rates of treatment avoidance and early discontinuation due to immune-related adverse events (IrAEs). In contrast, HPRR patients experienced greater clinical benefit and longer ICI exposure. CiRT classification was not associated with short-term imaging-based response outcomes, supporting its role as an independent predictor. Given that ICI therapy and supportive care can cost more than $850,000 per patient, CiRT offers potential value in avoiding ineffective treatment and associated toxicities. **Conclusions:** CiRT demonstrates meaningful clinical utility as a non-invasive, predictive tool for guiding immunotherapy decisions across tumor types. It enables more precise treatment selection, improves patient outcomes, and supports value-based cancer care.

## 1. Introduction

The rapid progression of novel immune-based treatment strategies in solid tumors, beginning with the approval of immune checkpoint inhibitors (ICIs) for melanoma in 2011 and subsequent approval of numerous agents in the ICI therapeutic category has revolutionized treatment of advanced cancers that were deemed invariably fatal until the past decade [1,2]. Faster development of novel biomarker-driven targeted therapies has accelerated hope and optimism about converting advanced solid tumors into chronic diseases. These advances have also led to the recognition of other clinical challenges, including unforeseen observations and complications associated with choosing the sequencing of appropriate therapeutic interventions as well as the management of drug-related complications.

A major challenge in oncology is selecting the most appropriate therapeutic intervention when broad-based profiling reveals multiple actionable mutations, targetable genomic alterations, or markers related to immunotherapy response [3]. Conversely, the absence of such markers also introduces clinical uncertainty and can limit treatment options.

Blood-based tools that reflect the host immune environment and inform treatment decisions across tumor types are urgently needed, particularly when tumor tissue is inaccessible, invasive procedures are not feasible, or biomarker results are delayed. Even when tissue is available, traditional tests often yield non-informative or inconclusive results.

The treatment landscape for metastatic cancer has been transformed by the advent of immunotherapy [4]. Multiple subclasses of immune checkpoint inhibitors (ICIs) are now approved, with the most widely used agents targeting cytotoxic T-lymphocyte-associated protein 4 (CTLA-4), programmed cell death receptor 1 (PD-1), and programmed death-ligand 1 (PD-L1) [5,6,7,8,9]. Currently, approximately five FDA-approved ICIs, primarily PD-1 and PD-L1 inhibitors, account for 99 percent of immunotherapy use in oncology [1]. These agents have demonstrated improved overall survival in a range of malignancies, including melanoma, non-small cell lung cancer (NSCLC), renal cell carcinoma, head and neck squamous cell carcinoma (HNSCC), urothelial carcinoma, and endometrial stromal sarcoma [10].

The past year has brought major advances in immune checkpoint inhibitor (ICI) therapy for solid tumors. The FDA recently approved a subcutaneous formulation of nivolumab, improving accessibility across multiple indications [11]. Pembrolizumab received expanded approval in HER2-positive gastric and gastroesophageal cancers based on the KEYNOTE-811 trial [12], and the nivolumab–ipilimumab combination was approved as first-line therapy for unresectable hepatocellular carcinoma following the CheckMate-9DW trial [13]. Collectively, these developments underscore the central and expanding role of ICIs in modern cancer care. Despite these advances, immuno-therapy is not a panacea and presents the following three challenges: 1) variability of response by tumor type and generally low objective response rate (ORR) with mono-therapy, 2) unreliable tools to predict patient response before therapy and 3) immuno-therapy-related side effects [14]. ICI ORRs vary significantly across tumor types and according to Zhao et al., PD-(L)1 only produced an ORR greater than 30% in three cancer subtypes: Lymphoma, Merkel cell cancer and Cutaneous cancer [15]. Traditional tools for identifying ICI candidates such as assessing PD-L1 expression in tumor biopsies using IHC have correlated poorly to patient outcomes [15]. Numerous studies have documented clinical responses to immune checkpoint inhibitors (ICIs) even in tumors classified as PD-L1–negative, highlighting the marker’s limited predictive value [1,16,17]. Moreover, traditional tissue biopsies offer only a narrow, static glimpse of tumor biology, often failing to reflect the full heterogeneity and temporal evolution of the disease [18]. As tumors adapt and progress, PD-L1 expression can change significantly over time, complicating efforts to accurately assess immunotherapy responsiveness from a single biopsy specimen [19]. Tumor mutation burden (TMB), and mismatch re-pair (MMR/MSI) analysis have shown limited utility and only in specific indications [20,21]. The need for a more accurate predictor of response to ICIs and standardization of patient screening modalities across indications is well-described in the literature.

Immunotherapy side effects can be severe or even deadly and at least mild Immune-related Adverse Events (IrAE) occur in 30–50% of patients [22]. Additionally, a small subset of patients on immunotherapy may not only fail to benefit from immunotherapy, instead, they experience a faster and more aggressive progression of the tumor than expected, with a dramatic acceleration of the disease, which is referred to as hyperprogression (HP) or hyperprogressive disease (HPD) [23,24,25,26]. Patients with HPD could suffer a deleterious survival effect and significantly shorter OS, suggesting that HP should be managed as fulminant toxicity and needs to be considered before immunotherapy is initiated [27,28].

Lastly, cost is another consideration that is becoming exceedingly more important with the rising price of drugs and the increased use of combination therapies. The treatment of a patient with an ICI and supportive care can reach a cost of $850,000 per patient [29,30]. To address health disparities and afford access, we must develop novel and accurate tools to help predict which patients are likely to benefit or, as important, those that will not benefit from these revolutionary therapies. The goal is to deliver personalized care to the patient maximizing clinical efficacy, while minimizing harm and effectively managing cost.

The EpiSwitch Checkpoint Inhibitor Response Test (CiRT) is an innovative blood-based assay designed to predict a patient’s likelihood of responding to immune checkpoint inhibitors (ICIs) [31]. Rather than relying on tumor tissue, CiRT analyzes epigenetic biomarkers, specifically, chromatin conformational signatures, detected in circulating immune cells. These three-dimensional genomic structures reflect the regulatory architecture of the genome and are critical for maintaining gene expression patterns and cellular identity [32,33,34,35]. In immune cells, such epigenomic configurations play a central role in modulating immune function and responsiveness, making them a valuable target for predictive biomarker development [35].

## 2. Methods

### 2.1. Study Design

This is a prospective observational real-world evidence study designed to evaluate the clinical utility of the EpiSwitch^®^ CiRT blood-based assay in identifying a patients’ likelihood of responding to immune checkpoint inhibitors (Protocol #40393314.0). This interim analysis of the PROWES study was designed and reported in accordance with the STROBE (Strengthening the Reporting of Observational Studies in Epidemiology) guidelines for observational cohort studies (Figure 1). The study enrolls patients diagnosed with stage II, III and IV cancer and who are candidates and/or planned to receive immune check point inhibitors as a therapy at the initial visit or potentially at a subsequent visit were offered CiRT. The study is designed for up to 2000 patients and this is an interim readout of patient data as defined in the protocol. Patients enrolled received subsequent CiRT testing every 3 months regardless of treatment decisions. Patients were followed for up to six months. Treatment administered, disease-free survival, overall survival, stable disease, progressive disease, complete response, time to recurrence, physician questionnaires and patient-reported outcomes were recorded for the duration of the patient’s time in the trial. No interventions were required for study purposes, and data was collected through chart review in compliance with HIPAA regulations.

### 2.2. IRB or Ethics Approval

This interim readout of the prospective observational study was in accordance with Good Clinical Practice (GCP) guidelines, where applicable, the Declaration of Helsinki and US Food and Drug Administration (FDA) guidelines where applicable. Data protection and privacy regulations were strictly observed in the capturing, forwarding, processing, and storing of patients’ data. The informed consent was required, the study received Institutional Review Board (IRB) approval from WIRB-Copernicus Group IRB (Puyallup, WA, USA); Study WO#: 1-1852726-1.

### 2.3. Patient Population

A total of 205 cancer patients with stage II, III and IV cancer were included. Inclusion and exclusion criteria were assessed by the investigator and are included in Table 1.

Eligible patients were 18 years or older, reflecting the adult validation of the CiRT assay and the focus of this observational study on adult oncology practice. Pediatric patients were excluded due to differences in cancer biology, treatment regimens, and regulatory frameworks.

### 2.4. Assay Performance

The CiRT assay was performed in a CLIA-certified laboratory using validated SOPs. Analytical precision has been confirmed, with intra-assay CVs ranging from 1.8% to 5.5% and inter-assay CVs from 2.8% to 5.9%, all within the prespecified acceptance criterion of <6%. Pre-analytical variables were controlled through standardized phlebotomy protocols, EDTA blood collection, and validated shipping conditions. Sample stability has been verified for up to 28 days under ambient transport conditions. The CiRT assay has been independently validated in both the UK and US laboratories and has received approval from the New York State Department of Health Clinical Laboratory Evaluation Program (CLEP).

#### 2.4.1. Testing Procedure

The EpiSwitch CiRT assay is a blood-based test that evaluates 3D genomic conformational biomarkers to stratify patients by likelihood of responding to ICIs. The test leverages chromatin conformation signatures at immune-regulatory loci, previously identified as predictive of response to checkpoint inhibitors, and applies a probabilistic classification model to stratify patients as high or low probability responders. It is the first test of its type and can predict the probability of checkpoint inhibitor success with an accuracy of 85% and a negative predictive value of 97%. The clinical validity of the test has been proven in published literature (PMC10216232) [31]. Results are reported as binary outputs: High probability of response (HPR) or low probability of response (LPR). Testing was performed in a CLIA-certified laboratory (Chester, Virginia, OBD Inc. partner lab) under standard operating conditions and reimbursement code CPT PLA 0332U. This PLA code is specific to the CiRT blood-based test and enables standardized reimbursement by Medicare, Medicaid, and private payors in the U.S. In the United Kingdon, BUPA UK covers CiRT tests run by UKAS-accredited clinical laboratory (Oxford, OBD plc.). Test reports with the CiRT results were generated automatically, with calls independent of biopsy or imaging data.

#### 2.4.2. Data Collection and Outcome Measures

The PROWES study was designed to assess the real-world clinical utility of the EpiSwitch CiRT assay in guiding immunotherapy decisions across diverse solid tumors. The primary endpoint was the impact of CiRT results on treatment decision-making, captured through physician questionnaires administered before and after receipt of test results. Secondary utility measures (Table 2) included concordance between CiRT-predicted likelihood of response and observed clinical outcomes (CR, PR, SD, PD), as well as time from diagnosis or consultation to treatment initiation. The study also captured the proportion of patients who avoided or were de-escalated from immune checkpoint inhibitor (ICI) therapy based on low-probability CiRT results, including those taken off treatment early due to immune-related adverse events (IrAEs). Longitudinal utility was assessed by tracking CiRT classification shifts over time (e.g., HPRR to LPRR) and monitoring disease progression or response in correlation with CiRT status. Additional analyses examined the relationship between CiRT results and social determinants of health (SDoH), including race, ethnicity, and socioeconomic status. Finally, health economic outcomes were modeled to evaluate potential cost savings from avoided ICI use, reduced IrAE management, and overall treatment efficiency enabled by CiRT-guided decision-making.

Data collected included demographic data (including Social Determinants of Health), medical and family history, type of ICI, number of cycles and outcomes, laboratory results, Immune-related Adverse Events (IrAE) and imaging testing.

Patients were classified as CiRT High or Low using a pre-specified threshold established in prior validation studies. This cut-off was locked before analysis of the current dataset and was not recalibrated to the present cohort.

### 2.5. Statistical Analysis and Model Development

To explore associations between clinical variables and CiRT classification, we implemented a supervised machine learning pipeline. The cohort was randomly divided into a training set and a held-out test set to evaluate model generalizability. Feature preprocessing included normalization, handling of missing data, and exclusion of sparse or collinear variables.

Multicollinearity was assessed using a correlation matrix and variance inflation factors (VIF). Variables with VIF greater than 5 were flagged as collinear and reviewed prior to modeling. Predictive models were trained using regularized classification techniques, with hyperparameters optimized via cross-validation. Model performance was assessed on the test set using accuracy, sensitivity, specificity, and confusion matrices (Figure 2).

Feature importance was quantified using SHAP (SHapley Additive exPlanations) values, which provided interpretable insights into which clinical features most strongly influenced CiRT High vs. Low classification. The final model identified sex, comorbidity burden, disease stage, and treatment history as top predictors of CiRT result. All analyses were conducted using open-source statistical and machine learning tools in R (v4.4.1). The following R packages were utilized: caret (v6.0-94), car (v3.1-2), corrplot (v0.92), MASS (v7.3-61), SHAPforxgboost (v0.1.3), shapviz (v0.9.4), tidyverse (v2.0.0), and xgboost (v1.7.7.1).

## 3. Results

### 3.1. Patient Population and Data Overview

A total of 205 patients with advanced solid tumors were included in this interim analysis. Data were collected from both academic and community oncology centers, with 67% of patients treated in community settings. The PROWES Feature Compare Tool was used to evaluate associations between EpiSwitch CiRT results and key clinical variables, including sex, lesion measurability, baseline clinical response, and treatment decisions. At baseline, 53% of patients had measurable disease by RECIST v1.1 criteria, while the remainder were treated empirically due to limited biopsy access, indeterminate imaging, or urgent clinical scenarios.

### 3.2. Clinical Predictors of CiRT Classification

To assess whether real-world clinical features correlate with EpiSwitch CiRT test results, we applied machine learning models to a training cohort (n = 128) and validated predictions on an independent test cohort (n = 82). The final logistic model accurately stratified CiRT High and Low classifications, with the confusion matrix confirming generalizable performance.

SHAP (SHapley Additive exPlanations) analysis revealed that female sex, lower comorbidity burden, earlier stage at diagnosis, and use of immunotherapy in earlier lines of therapy were the strongest drivers of CiRT High classification. These associations were consistent across both training and test sets and were further validated by recursive feature elimination and GLMNET stability selection (Figure 3).

Visualizations of SHAP importance and log-odds scores showed that patients with more favorable clinical profiles were more likely to receive a CiRT High result, while later-stage disease and higher comorbidity burdens skewed toward CiRT Low. These findings reinforce the biological interpretability of CiRT calls and support their alignment with immune fitness indicators commonly observed in oncology practice.

To further elucidate the clinical interpretability of the model, SHAP log-odds plots were generated for two key predictors: line of therapy and surgical resection status. These visualizations revealed that patients undergoing first-line therapy and those with no prior resection had higher SHAP values for CiRT High classification (Figure 4). Stage II disease was associated with more favorable odds across both features. These relationships reinforce the role of early intervention and intact immune architecture in shaping favorable epigenetic signatures (Figure 4).

### 3.3. Linear Discriminant Analysis (LDA) of CiRT Classification

To further validate the biological and clinical relevance of CiRT classification, we applied a Linear Discriminant Analysis (LDA) model trained on all available baseline clinical variables, excluding subject ID, sex, and weight to avoid introducing bias or confounding effects. Subject ID carries no predictive value, and both sex and weight were deliberately excluded to prevent the model from capturing known correlates of immune fitness and to allow for an independent assessment of sex-based effects.

The resulting LD1 scores showed clear separation between patients classified as CiRT High and CiRT Low, indicating that CiRT calls reflect an aggregate clinical signature composed of multiple independent baseline factors. This supports the premise that CiRT captures a biologically meaningful distinction in systemic immune status rather than reflecting any single dominant feature (Figure 5).

Although sex was not included as a predictor, stratifying LD1 scores by both CiRT classification and gender revealed a notable difference in males, where CiRT High and Low patients showed stronger separation. A similar, though slightly less pronounced, pattern was seen in females. These findings suggest that the CiRT test identifies an underlying immune readiness phenotype that is consistent across patient subgroups and not simply driven by superficial clinical characteristics (Figure 5).

Taken together with the SHAP-based machine learning results, the LDA provides additional evidence that CiRT classification is aligned with real-world indicators of systemic immune function. These results strengthen the case for CiRT as a biologically grounded and clinically actionable tool for guiding immunotherapy decisions.

### 3.4. Impact of CiRT on Clinical Decision-Making

CiRT results influenced physician treatment decisions in 61% of cases, based on paired physician surveys completed before and after test result disclosure. In the low-probability response group (LPRR), 46% of patients were either not initiated on ICI therapy or had their immunotherapy de-escalated early, often due to rapid disease progression or emerging immune-related adverse events (IrAEs). Conversely, 74% of high-probability response patients (HPRR) had their treatment continued or escalated based on favorable CiRT status. Notably, CiRT enabled earlier alignment of treatment strategies with patient-specific immune readiness, particularly in tumors with limited biomarker guidance.

### 3.5. CiRT Result Distribution by Sex

A statistically significant association was observed between sex and CiRT classification (*p* = 0.0035). Among female patients (n = 104), 47 were classified as CiRT High (45.2%), while 57 were CiRT Low (54.8%). Among male patients (n = 101), only 25 were CiRT High (24.8%), and 76 were CiRT Low (75.2%). These findings demonstrate a nearly twofold higher prevalence of CiRT High status in females versus males, suggesting possible biologic sex differences in systemic immune readiness.

### 3.6. Lesion Measurability and CiRT Status

Lesion measurability exhibited a borderline significant association with CiRT classification (*p* = 0.0525). Of the 92 patients with measurable disease, 25 (27.2%) were CiRT High and 67 (72.8%) were CiRT Low. Among the 83 patients without measurable disease, 32 (38.6%) were CiRT High and 51 (61.4%) were CiRT Low. This trend may suggest that patients with measurable tumor burden exhibit more pronounced immune resistance features.

### 3.7. Baseline Clinical Response and Lesion Measurability

Baseline response categories significantly differed by lesion measurability (*p* = 0.0295). Patients with measurable lesions (n = 92) had the following response breakdown: Complete Response (CR): 3.3%, Partial Response (PR): 7.6%, Stable Disease (SD): 43.5%, Progressive Disease (PD): 5.4%, Unknown (UN): 40.2%. For those without measurable disease (n = 83): CR: 3.6%, PR: 3.6%, SD: 31.3%, PD: 3.6%, UN: 57.8%.

### 3.8. CiRT Result and Baseline Response

No significant association was found between CiRT result and baseline response (*p* = 0.9118). In the CiRT High group (n = 72), the response distribution was CR: 2.8%, PR: 6.9%, SD: 33.3%, PD: 1.4%, UN: 55.6%. Among CiRT Low patients (n = 133): CR: 3.0%, PR: 4.5%, SD: 34.6%, PD: 5.3%, UN: 50.4%. These findings reinforce the utility of CiRT as a prognostic tool rather than a real-time surrogate for radiographic response.

These findings demonstrate how the integration of the EpiSwitch CiRT test with clinical data, through structured tools such as the PROWES Group Compare interface, can offer oncologists deeper insight into treatment stratification, particularly in determining suitability for immunotherapy versus alternative approaches.

### 3.9. Longitudinal and Monitoring Utility

Serial CiRT testing was performed in a subset of patients (n = 15), revealing transitions from HPRR to LPRR in association with disease progression. While the epigenetic signals evaluated by CiRT are stable, they are conditional and reflective of the phenotype. As the phenotype of the cancer changes, so too will the CiRT result. This supports the potential of CiRT as a dynamic monitoring tool to detect evolving immune escape or resistance patterns during ICI therapy.

### 3.10. Health Equity and Access Assessment

CiRT results were analyzed in the context of race, ethnicity, and socioeconomic status (SES). No significant disparities were observed in the distribution of HPRR or LPRR classifications across demographic groups, suggesting CiRT offers equitable diagnostic utility in diverse populations.

### 3.11. System-Level Utility and Cost Considerations

Given that ICI treatment and associated supportive care costs can exceed $850,000 per patient in some advanced cancer scenarios, CiRT-guided avoidance of ineffective therapy represents a meaningful opportunity for cost containment. Even small adjustments, such as ending an ineffective ICI therapy one or two cycles early, will reflect a cost avoidance many multiples higher than the cost of the CiRT assay. Economic modeling is ongoing, but early projections support the integration of CiRT into value-based care pathways to reduce unnecessary toxicity, streamline treatment selection, and improve resource stewardship.

## 4. Discussion

### 4.1. Overview of Clinical Utility

In this study, we investigated the clinical relevance of the EpiSwitch CiRT blood test for stratifying patients being considered for immunotherapy versus other anticancer therapies. Our findings demonstrate that CiRT results are significantly associated with patient sex and lesion measurability, but not with short-term radiographic response outcomes. These insights support the potential utility of the CiRT assay as a tool to guide oncologic treatment decisions, particularly in situations where lesion status or treatment intent are heterogeneous.

### 4.2. Sex-Based Differences in CiRT Classification

The higher prevalence of CiRT High results in female patients (45.2%) compared to males (24.8%) suggests that underlying immunologic or epigenetic differences may contribute to sex-specific immune readiness. CiRT is designed as a pan-cancer assay that captures systemic immune profile settings relevant to PD-1/PD-L1 signaling, rather than tumor-specific features. Accordingly, the observed sex difference likely reflects host immune biology across cancers, not confounding effects from tumor distribution (e.g., breast versus prostate cancer). This interpretation is consistent with previously reported immune dimorphism across sexes, including differences in checkpoint gene expression, cytokine regulation, and epigenomic configuration [36]. Females also exhibit stronger baseline immune activation and higher prevalence of autoimmune diseases such as rheumatoid arthritis, which is among the most frequent immune-related adverse events during checkpoint inhibitor therapy [37]. Taken together, the nearly twofold higher prevalence of CiRT High classification in females may reflect these broader sex-based differences in systemic immunity. While compelling, this observation should be considered hypothesis-generating and warrants further validation in disease-specific cohorts in future studies.

### 4.3. Lesion Measurability and Immune Readiness

A greater proportion of patients with measurable lesions (72.8%) were classified as CiRT Low, suggesting a potential link between tumor burden and reduced immunologic compatibility for ICI therapy. The observed inverse relationship between lesion quantifiability and CiRT High status supports the hypothesis that more extensive disease may suppress systemic immune capacity in ways detectable by 3D genomic profiling [38]. This association has clinical utility across several dimensions.

First, it supports the use of CiRT as a triage tool to help identify patients who may be less likely to benefit from ICIs upfront, particularly in advanced disease where timing and immune function are critical. Second, it reinforces the idea that CiRT captures a systemic signal that may reflect the immunologic impact of tumor burden, which is not often accounted for in proteomic or genomic biomarker frameworks. Lastly, the finding provides biologic validation that the CiRT test reflects meaningful immune readiness shaped by both host factors and disease extent, strengthening confidence in its clinical relevance.

### 4.4. Role of Structured Analytics in Real-World Settings

The use of the PROWES Feature Compare tool in this study allowed efficient, re-producible subgroup analyses, enhancing our ability to detect clinically relevant associations in real-world patient data. The ability to rapidly explore stratifications by CiRT result, lesion status, and therapy history enhances the translational potential of this assay in clinical practice.

While this study is limited by the differences in treatment regimens and various cancerous indications and the relatively high proportion of unknown response outcomes, the insights generated lay a foundation for further prospective validation. Larger studies incorporating longitudinal outcomes, progression-free survival, and treatment durability metrics will be necessary to fully elucidate the predictive and prognostic value of the EpiSwitch CiRT assay in diverse cancer populations.

### 4.5. Clinical Integration and Decision Impact

A key value proposition of the EpiSwitch CiRT test lies in its ability to influence real-time treatment decisions, especially when traditional biomarkers are absent, indeterminate, or logistically impractical. In many cases, physicians are faced with a therapeutic crossroad, choosing between immune checkpoint inhibitors (ICIs) or alternative options such as chemotherapy or tyrosine kinase inhibitors. The CiRT test introduces a non-invasive, binary framework to help navigate that decision with confidence.

For example, in hepatocellular carcinoma or metastatic bladder cancer, where both immunotherapy and targeted agents are considered standard, CiRT can be used to triage patients toward the most appropriate first-line regimen based on predicted immune compatibility. Similarly, in community oncology settings, the rapid turnaround and no tissue requirements for CiRT make it highly compatible with current workflows. It allows oncologists to avoid defaulting to ICIs in ambiguous cases, and instead, use evidence-based stratification to guide care.

This decision support function aligns closely with modern oncology’s emphasis on personalization, resource stewardship, and rapid initiation of effective treatment. As such, CiRT represents not just a biomarker but a workflow optimization tool.

### 4.6. Validation in Hepatocellular Carcinoma and GI Tumors (Ouf et al.)

Further evidence of CiRT’s predictive relevance comes from a retrospective cohort study by Ouf et al. (also presented as He et al.), which evaluated immunotherapy outcomes in 43 patients with hepatocellular carcinoma (HCC) and other gastrointestinal malignancies [39]. Patients classified as high probability (HP) responders by CiRT experienced significantly better outcomes than their low probability (LP) counterparts. Notably, the median progression-free survival (PFS) for the LP group was only 2.0 months, while median PFS in the HP group was not reached during the study period (p = 0.044). Additionally, 70.8% of HP patients achieved clinical benefit (CR, PR, or SD) compared to only 31.6% in the LP group (p = 0.0098). These findings align with our current analysis and emphasize CiRT’s potential to guide treatment decisions in real-world oncology practice. Importantly, the observed PFS benefit in HCC underscores the clinical utility of CiRT in a tumor type where ICI response is variable and predictive biomarkers are urgently needed.

The clinical workflow in this study (Figure 6) reflects the underlying hypothesis of Ouf et al., which proposes using the EpiSwitch CiRT test to stratify patients with unresectable hepatocellular carcinoma (uHCC) based on immunologic readiness prior to first-line systemic therapy. By integrating CiRT classification into existing Child-Pugh and ECOG-based treatment pathways, the approach aims to optimize immunotherapy selection and avoid ineffective or harmful treatments in vulnerable liver cancer populations. This strategy may also help gastrointestinal oncologists select non-inferior therapeutic options that maximize progression-free and overall survival while minimizing toxicity risk. This is especially critical in uHCC, where approximately 70 percent of uHCC patients do not proceed to second-line therapy due to rapid clinical decline or treatment-related toxicity.

Complementing these findings, a forthcoming study by He et al. (Georgetown University Medical Center and the Herbert Irving Comprehensive Cancer Center at Columbia University) will report overall survival and progression-free survival outcomes in HCC patients treated with ICIs, stratified by CiRT classification, thereby providing retrospective outcome concordance for the proposed decision-tree framework. Notably, interim data from this study presented at ASCO GI 2025 demonstrated promising results, suggesting that CiRT can help optimize survival in unresectable HCC patients by informing and refining first-line therapy selection [39].

### 4.7. Comparison with Standard Biomarkers

Traditional markers used to predict ICI response, including PD-L1 expression, tumor mutational burden (TMB), and microsatellite instability (MSI), each suffer from limited sensitivity, variable inter-lab reproducibility, and challenges with spatial and temporal heterogeneity. PD-L1 demonstrates discordant correlation with outcomes across tumor types, and its utility is often diminished in real-world scenarios by tissue availability constraints or assay variability.

In contrast, the CiRT assay offers a tumor-agnostic, blood-based approach that captures the patient’s systemic immunologic architecture rather than focusing solely on the tumor microenvironment. This distinction may explain why CiRT has shown strong negative predictive value in prior validation work and appears to capture information orthogonal to traditional genomic or IHC-based tools. From a clinician’s perspective, CiRT’s simplicity and cross-indication applicability reduce diagnostic ambiguity and allow broader, more equitable deployment.

### 4.8. Evidence from Prospective Studies

Evidence supporting the clinical relevance of CiRT continues to emerge from prospective trial settings. In the recently published JAVELIN Bladder 100 trial analysis, CiRT stratification aligned with tumor immune infiltration and correlated with survival outcomes in patients receiving avelumab maintenance immunotherapy [33]. Importantly, these data were derived from a Phase 3 trial cohort and underscore CiRT’s potential to function not only as a predictive biomarker but also as a companion tool for immune profiling, particularly when tumor biopsy is impractical.

The ability to extract meaningful stratification from peripheral blood samples positions CiRT as a unique and scalable tool in both academic and community practice settings. These results complement real-world observations from this interim analysis, reinforcing CiRT’s clinical utility across both controlled and uncontrolled environments.

### 4.9. Future Utility and System-Level Impact

Looking forward, the CiRT test holds promise beyond individual treatment decisions. Its adoption could improve system-level efficiency by reducing time-to-treatment initiation, minimizing patient exposure to ineffective regimens, and lowering the burden of managing immune-related toxicities. In settings with limited access to tissue-based diagnostics or where rapid triage is essential, CiRT offers a practical, reproducible, and evidence-backed solution.

Beyond its impact on clinical decision-making and cost-efficiency, the CiRT assay is well-suited to decentralized care and teleoncology models. Its blood-based format, absence of tissue requirements, and rapid turnaround make it easy to integrate into community oncology practices, remote infusion centers, and rural clinics where traditional biomarker testing is often limited or delayed [34]. As oncology care increasingly adopts virtual consultations, shared decision-making, and hub-and-spoke delivery systems, CiRT’s simplicity and tumor-agnostic design position it as a practical tool to extend precision immunotherapy guidance beyond academic centers. By enabling access in lower-resource settings, CiRT also advances equity in cancer care, ensuring that biomarker-informed decisions are not confined to patients treated at major institutions [36].

As value-based models of oncology care evolve, tools like CiRT which are simple, informative, and actionable are poised to play a defining role in the next generation of precision cancer care.

### 4.10. RECIST Correlation (Interim Dataset)

The lack of correlation observed between CiRT and RECIST (*p* = 0.9118) likely reflects the immaturity of this interim dataset. RECIST is designed to capture changes in tumor size, which may not directly align with the immunologic dynamics that CiRT measures. The observed associations with sex (*p* < 0.0001) and lesion measurability (*p* = 0.0295) suggest that CiRT captures biologic variability beyond structural imaging endpoints. As additional follow-up accrues, larger datasets will allow for a more complete evaluation of potential correlations between CiRT and RECIST-defined outcomes.

Interestingly, the higher proportion of CiRT High classification in females compared to males (45.2% vs. 24.8%) is consistent with established sex-based immune dimorphism, where women often exhibit stronger T-cell activity and enhanced innate immune responses [40]. Hormonal status and sex chromosome-linked differences in immune gene regulation likely contribute to these findings, although these variables were not captured in this interim analysis. These observations underscore the need for future studies to explore sex-specific immune modulation in relation to CiRT. Mechanistically, the CiRT algorithm reflects systemic immune readiness by capturing 3D genomic signatures of T-cell regulation, interferon signaling, and checkpoint pathway activity, which collectively distinguish patients more likely to derive benefit from immune checkpoint blockade. However, beyond RECIST correlation, other dataset-specific limitations also warrant consideration.

### 4.11. Impact of Unknown Response Outcomes

A notable limitation of this interim dataset is the high proportion of cases categorized as “Unknown” with respect to treatment response. To address this, we performed sensitivity analyses excluding these cases, which confirmed that the predictive performance of CiRT remained consistent with the primary analysis. While the presence of “Unknown” outcomes reflects the real-world heterogeneity of follow-up and documentation, we recognize that this may introduce bias and emphasize the need for further validation in larger, prospectively monitored cohorts that will continue to be analyzed throughout this study.

### 4.12. Absence of Direct Biomarker Comparisons

A limitation of this interim analysis is the absence of a robust head-to-head comparison between CiRT and established biomarkers such as PD-L1 expression, tumor mutational burden (TMB), or microsatellite instability (MSI). Although some data are available, these ancillary biomarkers were not uniformly captured across tumor indications in this heterogeneous real-world cohort, reflecting the lack of standardization of TMB and MSI testing in many solid tumors. As a result, our benchmarking necessarily relied on published performance metrics. Ongoing and future prospective studies are designed to incorporate standardized parallel biomarker profiling, enabling direct ROC-based comparisons and a clearer positioning of CiRT relative to other biomarkers.

### 4.13. Limitations of the Longitudinal Subset

While the longitudinal subset (n = 15) provided preliminary evidence that CiRT may track treatment-related immune changes over time, the small sample size precludes firm conclusions. These data should be viewed as hypothesis-generating, and additional prospective studies with larger longitudinal cohorts will be required to validate the utility of CiRT as a dynamic monitoring tool.

### 4.14. Lack of Mature Outcomes in CiRT-Influenced Decisions

A limitation of this interim report is that long-term outcomes among the 61% of patients whose treatment decisions were influenced by CiRT are not yet available. As follow-up matures, future analyses will evaluate response rates and survival outcomes in this subgroup, providing additional evidence of CiRT’s clinical impact on treatment efficacy.

### 4.15. Analytical Validity and Reproducibility

Despite these limitations, the study provides strong real-world evidence of CiRT’s clinical impact. CiRT’s analytical performance has been validated across two independent laboratories, with repeatability (intra-assay CVs 1.8–5.5%) and reproducibility (inter-assay CVs 2.8–5.9%) consistently meeting acceptance criteria. CLEP approval provides additional regulatory assurance of its high fidelity in clinical practice. The reproducibility of the CiRT assay is further supported by the use of a pre-specified High/Low threshold that has been applied consistently across independent cohorts. This same cut-off was validated in the Phase 3 JAVELIN Bladder 100 trial conducted in collaboration with Pfizer, where CiRT classification correlated with immune-related treatment outcomes [33]. Taken together, these findings support both the technical reliability and the clinical utility of CiRT.

### 4.16. Concluding Insights on Clinical Impact

These findings reflect not only the predictive performance of the CiRT test, but also how physicians and care teams adapt their treatment strategies when armed with a tumor-agnostic, blood-based tool that stratifies immune readiness. The real-world setting of the PROWES study illustrates how CiRT integrates seamlessly into clinical workflows, influencing decisions across diverse cancer types, regardless of traditional biomarker availability. This behavioral shift underscores CiRT’s role in enabling more consistent, biology-driven care across the oncology spectrum.

## 5. Conclusions

Immune checkpoint inhibitors continue to expand their role in the treatment of solid tumors. New formulations and expanded approvals reflect broadening clinical utility, while advances in immuno-oncology highlight the potential for transformative outcomes [11,12,13]. Within this evolving landscape, our findings support the value of CiRT as a non-invasive tool to guide immunotherapy decision-making and align patient care with the latest advances in the field.

This interim analysis of the PROWES study, a large prospective real-world evidence investigation, demonstrates that the EpiSwitch^®^ CiRT blood test provides meaningful and actionable clinical utility for guiding immunotherapy decisions across a broad spectrum of advanced solid tumors. Unlike PD-L1 immunohistochemistry, tumor mutational burden (TMB), or mismatch repair/microsatellite instability (MMR/MSI) testing, which have shown variable predictive performance and are limited by tumor type, biopsy access, and genomic complexity, CiRT delivers a universally applicable, blood-based classifier with a binary output that is easy to interpret and implement in routine oncology practice.

CiRT classifications were concordant with clinical outcomes. Patients with low-probability response results (LPRR) were more likely to experience disease progression, early discontinuation of immune checkpoint inhibitors (ICIs), and reduced clinical benefit. In contrast, high-probability response results (HPRR) were associated with higher rates of objective responses and durable disease control. Notably, patients with HPRR who received ICIs in the first- or second-line setting demonstrated particularly favorable outcomes, highlighting the importance of aligning immune readiness with early intervention.

CiRT results also directly influenced physician decision-making, including the avoidance of ICI therapy in LPRR cases and earlier de-escalation in the setting of immune-related adverse events (IrAEs). Additionally, the test proved useful in treatment landscapes with non-inferior therapeutic alternatives, such as ICIs versus tyrosine kinase inhibitors (TKIs) in unresectable hepatocellular carcinoma, helping clinicians tailor decisions based on patient-specific immunologic fitness.

Importantly, this study also captures how physicians adjust clinical decision-making when given access to a universal, blood-based predictor of ICI response. Unlike tumor-specific biomarkers, CiRT enables consistent, actionable insights across solid tumor types, helping to standardize and personalize care pathways in real-world oncology practice.

From a systems-level perspective, CiRT offers a compelling cost-effectiveness proposition. With the total costs of ICI therapy and supportive management exceeding $850,000 per patient in certain clinical scenarios, the ability to avoid or prematurely end an ineffective treatment has significant financial and resource implications. By guiding therapy away from patients unlikely to benefit, CiRT reduces unnecessary toxicity, shortens time to more appropriate alternatives, and supports value-based oncology care. Furthermore, exploratory analyses suggest utility in longitudinal surveillance, resistance monitoring, and equity assessment through stratification by social determinants of health (SDoH). The CiRT result is reflective of the tumor phenotype and the patient’s immune system, which will remain stable until there is a change to the phenotype. As adoption expands and longer-term outcomes are tracked, CiRT is positioned to become a critical tool in precision immuno-oncology. It improves the alignment of therapies with biological likelihood of benefit, enhances patient care, and supports more sustainable and evidence-driven cancer treatment strategies.

## Figures and Tables

**Figure 1 cancers-17-02900-f001:**
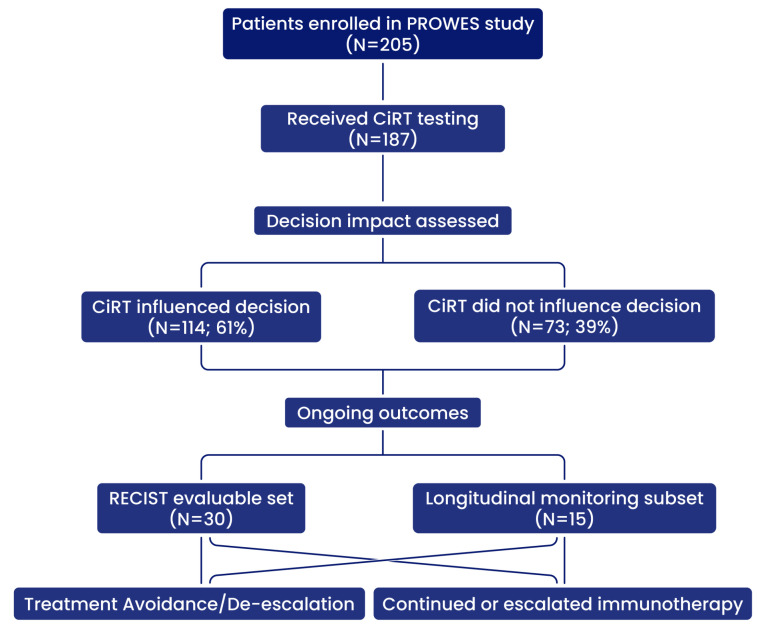
Study schema for patient enrollment, CiRT testing, decision impact, and evaluable outcome subsets.

**Figure 2 cancers-17-02900-f002:**
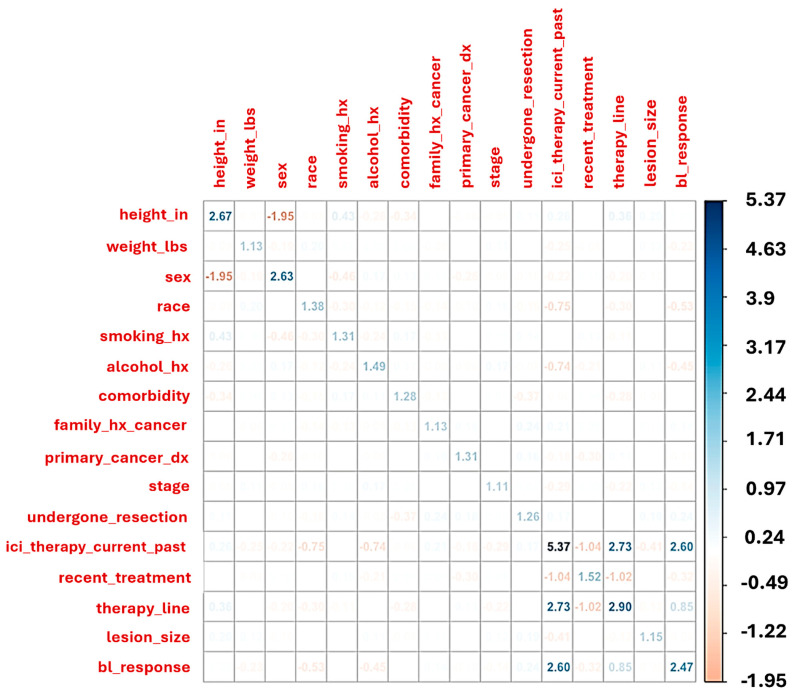
This heatmap displays pairwise correlations and variance inflation factors (VIF, on the diagonal) across 205 patient baseline features. Strong negative correlation is observed between sex and height, while ICI therapy status (current or past) shows collinearity with line of therapy and baseline response classification. Variables with VIF > 5, such as “ICI therapy current/past,” suggest moderate-to-high collinearity and were evaluated for exclusion or adjustment in multivariate modeling to reduce redundancy and overfitting.

**Figure 3 cancers-17-02900-f003:**
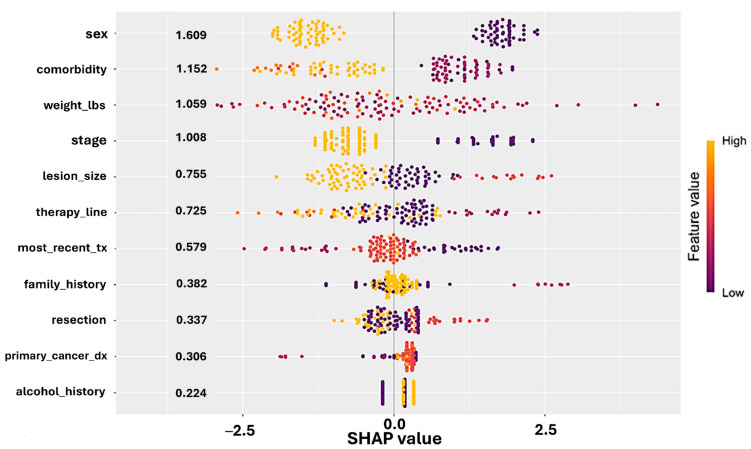
SHAP (SHapley Additive exPlanations) values derived from a logistic regression model predicting CiRT classification. Each dot represents a patient, with position on the *x*-axis indicating the impact (SHAP value) of that feature on the model output. Features are ranked by average importance. Color denotes the actual feature value (yellow = high, purple = low). Female sex, lower comorbidity burden, earlier cancer stage, and earlier line of therapy were associated with CiRT High classification. These results support the biological plausibility and clinical alignment of CiRT calls with real-world indicators of immunologic fitness.

**Figure 4 cancers-17-02900-f004:**
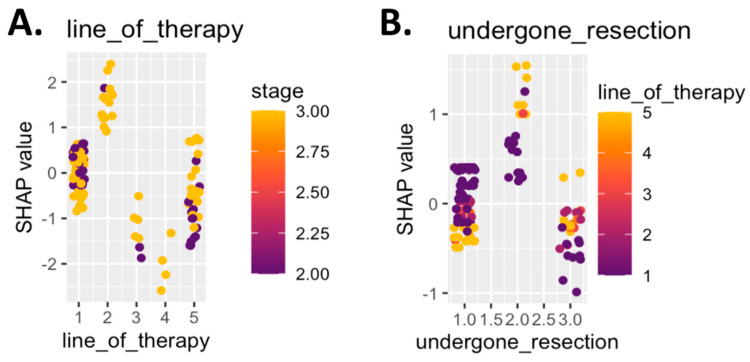
SHAP log-odds plots for top predictive clinical features. Panel (**A**) shows SHAP values for line of therapy, colored by cancer stage. Patients receiving first-line therapy with stage II disease showed the strongest association with a CiRT High classification, while later lines of therapy and stage III disease were more often associated with a Low classification. Panel (**B**) displays SHAP values for resection status, colored by line of therapy. Patients who had not undergone prior surgical resection and were receiving first-line therapy were more likely to be classified as CiRT High. Higher SHAP values indicate greater model contribution toward a High classification, reinforcing the role of clinical indicators of immune readiness in predicting likelihood of response to ICI therapy.

**Figure 5 cancers-17-02900-f005:**
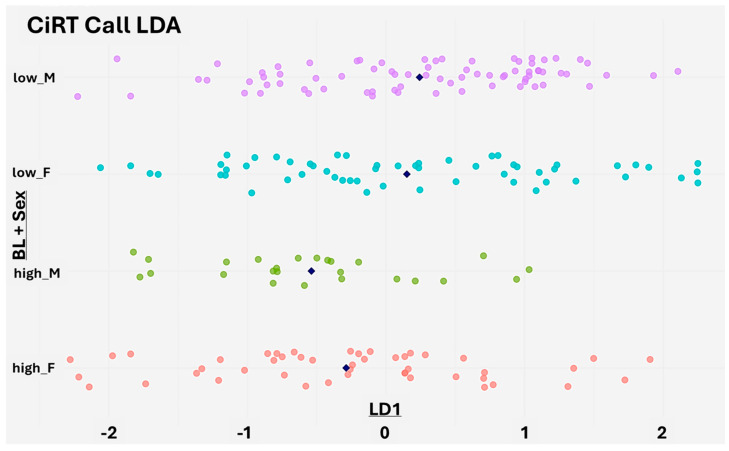
Linear Discriminant Analysis (LDA) of baseline clinical features stratified by CiRT classification and sex. The *X*-axis shows LD1 scores from an LDA model trained to predict CiRT classification using all available baseline clinical variables, excluding subject ID, sex, and weight. The *Y*-axis groups individuals by a combination of CiRT call (high or low probability of response) and sex, which was not used in model training. Each dot represents a patient, and navy rhombus indicate the group mean. Clear separation between CiRT High and Low groups is observed, particularly among male patients, suggesting that CiRT captures a composite immune readiness signal derived from the full clinical profile.

**Figure 6 cancers-17-02900-f006:**
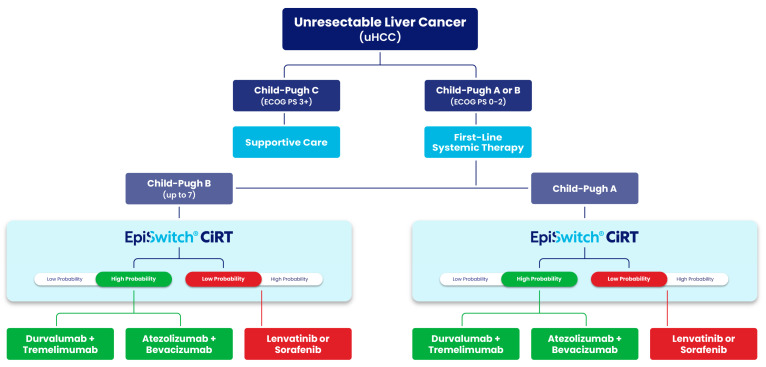
Treatment decision algorithm for patients with unresectable hepatocellular carcinoma (uHCC) incorporating EpiSwitch^®^ CiRT results. EpiSwitch CiRT results help stratify patients by probability of response to immune checkpoint inhibitors (ICIs). High-probability CiRT results support selection of ICI-based regimens such as durvalumab + tremelimumab or atezolizumab + bevacizumab, while low-probability results may prompt consideration of kinase inhibitors (lenvatinib or sorafenib). Adapted from clinical decision modeling aligned with findings from Ouf et al., where CiRT status was associated with progression-free survival and treatment response in uHCC patients receiving ICIs.

**Table 1 cancers-17-02900-t001:** Inclusion and Exclusion Criteria for Enrollment in the PROWES Study. Eligible patients were adults with advanced (Stage II, III or IV) solid tumors, deemed clinically appropriate for immune checkpoint inhibitor (ICI) therapy and selected by their provider to receive the EpiSwitch CiRT^®^ test as part of standard care. Key exclusion criteria included pregnancy, history of organ or bone marrow transplant, contraindication to ICI therapy, or serious medical conditions impairing participation.

**Inclusion Criteria**	1. 18 years of age or older 2. Stage II, III or IV cancer 3. Selected by their healthcare provider to receive the EpiSwitch CiRT test according to the current evidence-based schedule (per protocol) as part of their standard of practice. 4. ECOG performance status ≤ 2 5. Clinically eligible for ICI therapy 6. Able to read, understand and provide written informed consent. 7. Willing and able to comply with the study requirements
**Exclusion criteria**	1. Pregnant or breastfeeding 2. History of bone marrow or organ transplant 3. Contraindication for receiving Immune Check Point inhibitor 4. Serious medical condition that may adversely affect ability to participate in the study

**Table 2 cancers-17-02900-t002:** Primary and secondary clinical utility endpoints assessed in the PROWES study, spanning decision-making, workflow efficiency, treatment optimization, longitudinal monitoring, equity, and economic value.

Outcome	Utility Domain	Measured As
Change in treatment decision	Decision support	Physician-reported pre-/post-CiRT result
Concordance of LPRR to no response	Clinical validity and utility	Radiographic response (RECIST v1.1)
Treatment avoidance	Cost and toxicity mitigation	% of LPRR patients not treated with ICI
Early discontinuation of ICI	Toxicity mitigation	% of LPRR patients de-escalated from ICI
Time to ICI start	Workflow efficiency	Days from diagnosis to treatment start
SDoH stratification	Equity and access	CiRT status vs. race/ethnicity/SES
Resistance detection	Longitudinal monitoring	Proportion shifting from HPRR to LPRR
HEOR value	Economic utility	Modeled cost savings from CiRT guidance

## Data Availability

The data supporting the findings of this study are available from the corresponding author upon reasonable request. Due to patient privacy and institutional policies, individual-level clinical data are not publicly accessible.

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
