# Peer review of "Clinical Utility of the EpiSwitch CiRT Test to Guide Immunotherapy Across Solid Tumors: Interim Results from the PROWES Study"

_cancers, 2025, doi:10.3390/cancers17172900_

Round 1

Reviewer 1 Report

Comments and Suggestions for Authors

Major Revision:

While the study demonstrates promising clinical utility of the EpiSwitch CiRT test, several limitations warrant clarification. The high proportion of "Unknown" response outcomes (40.2–57.8%) may bias the analysis; sensitivity analyses excluding these cases are needed. The lack of correlation between CiRT and RECIST (p=0.9118) requires deeper mechanistic explanation, particularly given the observed sex-based (p<0.0001) and lesion measurability (p=0.0295) associations. Additionally, the small longitudinal subset (n=15) limits conclusions about dynamic monitoring utility. Address these gaps with expanded statistical validation and prospective cohort data.

  1. The manuscript lacks details on CiRT's analytical validity. What are inter-/intra-assay CVs? How was pre-analytical variability (e.g., blood collection timing, shipping conditions) controlled?
  2. The 45.2% vs. 24.8% CiRT High in females vs. males warrants deeper biological exploration. Were hormonal status or sex chromosome effects considered?
  3. How was the High/Low CiRT threshold determined? Was it pre-specified or data-derived? This affects reproducibility.
  4. Figure 5's decision tree lacks validation data. What was the observed outcome concordance when applying this algorithm retrospectively?
  5. The PD-L1/TMB/MSI comparison is superficial. Show direct comparison data (e.g., ROC curves) in the same patient cohort rather than literature references.

Reviewer 2 Report

Comments and Suggestions for Authors

The manuscript “Clinical Utility of the EpiSwitch CiRT Test to Guide Immunotherapy Across Solid Tumors: Interim Results from the PROWES Study” by Abdo et al. presents an interim analysis of 205 patients with advanced solid tumors enrolled in a prospective real-world evidence study. The primary endpoint measured how CiRT results changed physicians’ treatment decisions by comparing pre- and post-test surveys. Secondary endpoints included treatment avoidance, time to initiation of immune checkpoint inhibitor therapy, concordance between CiRT predictions and actual clinical responses, early discontinuation rates, and an exploratory health-economic model. Additional analyses explored longitudinal CiRT utilization, resistance monitoring, and equity across social determinants of health. In most cases, CiRT results shaped clinical decision-making, demonstrating meaningful clinical utility as a non-invasive, predictive assay that enables more precise immunotherapy selection, improves patient outcomes, and advances value-based cancer care.

The findings are interesting and appreciable, manuscript has little scope to improve further in the reference section only. Taking all points into my account, I’m recommending to accept the manuscript after addressing the following point.

Specific comment

Minor Point –

  1. Authors missed some “immune checkpoint and cancer immunotherapy” related resent time important discoveries. They need to pay attention in the “references” section of the manuscript.

Reviewer 3 Report

Comments and Suggestions for Authors

In this observational study authors investigated Clinical Utility of the EpiSwitch CiRT Test to Guide Immuno- 2 therapy Across Solid Tumors: Interim Results from the 3 PROWES Study. 

The manuscript is well written and following are some comments. 

  1. Since CiRT is highly personalized diagnosis author should provide some insight about the mechanistic parameters involved in its algorithmic calculation to identify positive vs negative. 
  2. Why the cut of age was 18 years in this observational study?
  3. Authors mentioned that 61% decision were influenced by CiRT. It would be ideal to provide a consort diagram showing how many patients were enrolled in this observational study and how many of them were consider by the end of the study for. 
  4. Among 61% patients, did author have a follow up of how many of them responded to the checkpoint therapy vs non responder.

Round 2

Reviewer 1 Report

Comments and Suggestions for Authors

Accept in present form

Author Response

Thank you for the reviewer for recommending our paper for publication. We appreciate the work you put in making this manuscript better. 

Reviewer 3 Report

Comments and Suggestions for Authors

The authors have addressed all the concerns and I have no further question. 

Author Response

Thank you for recommending out paper for publication. We appreciate all of the work you put into making this manuscript better.